# An In-depth Study of Bandwidth Allocation across Media Sources in Video Conferencing

## ABSTRACT

Video Conferencing Applications (VCAs) are indispensable for real-time communication in remote work and education by enabling simultaneous transmission of audio, video, and screen-sharing content. Despite their ubiquity, there is a noticeable lack of research on how these platforms allocate network bandwidth resources, especially under limited conditions, and how these resource allocation strategies affect the users' Quality of Experience (QoE). This paper addresses this research gap by conducting an in-depth analysis of bandwidth allocation strategies among prominent VCAs, including Zoom, Webex, and Google Meet, with an emphasis on their implications for QoE. To assess QoE effectively, we propose a general QoE prediction model based on data collected from a user study involving 800 participants. This study marks a pioneering effort in the extensive evaluation of multimedia transmissions across diverse media source scenarios and network conditions for VCAs and represents a significant advancement over prior research that predominantly concentrated on the quality assessment of singular media types. The promising outcomes highlight the model's effectiveness and generality in accurately predicting QoE across various scenarios among VCAs.

## 1 INTRODUCTION

To enhance telepresence, video conferencing platforms have gradually integrated various media sources, including audio, video from camera streams, screen from screen-sharing streams, chat, and other advanced functionalities. This integration of multimedia transmission facilitates a highly customizable communication experience, enabling users to dynamically select and modify media inputs to suit their specific virtual meeting requirements. In real-world video conferencing, users often use multiple media sources simultaneously. For example, during online classes, teachers may use audio, video, and screen-sharing media sources simultaneously to provide their students with a comprehensive and enriching learning experience.

Some prior studies analyzed the performance of popular VCAs [5, 16, 24, 29] and revealed their designs, including QoE metrics, network utilization, congestion control, etc. Others introduced innovative frameworks [4, 32] or systems [6] to enhance QoE. However, these studies mainly focused on individual media sources. There is still a significant research gap in exploring bandwidth allocation

*ACM MM, 2024, Melbourne, Australia*

across different media sources within VCAs, which is an essential factor for optimizing performance and user satisfaction in video conferencing environments.

Under bandwidth constraints, video conferencing quality could degrade without careful resource allocation. The overall QoE depends on the combined performance of all concurrent multimedia sources. In scenarios with restricted bandwidth, how to allocate bandwidth—whether prioritized for one media source, divided equally among all media sources, or distributed unevenly—becomes critical in determining QoE. For example, in an online class with limited bandwidth, allocating all bandwidth to support screen-sharing clarity while ignoring audio transmission may make it difficult for students to follow the screen-sharing contents without clear audio, resulting in a reduced QoE. Instead, if each media source receives a proportionate share of the bandwidth to function at an acceptable level of quality, the overall QoE could be considerably enhanced. Therefore, investigating bandwidth allocation strategies that balance different media sources within network constraints is vital for optimizing the overall QoE.

To address this, we perform in-depth measurement and modeling of bandwidth allocation for three media sources: audio, video (camera streams), and screen (screen-sharing streams). Our study begins by examining the bandwidth allocation strategies of three major commercial VCAs: Zoom, Webex, and Google Meet. Specifically, we focus on Zoom to examine its bitrate adaptation for each media source individually. Following this preliminary analysis, we conduct a broad user study to (1) explore the impact of different bandwidth allocation strategies on the QoE for real users and (2) develop a QoE prediction model general to various VCAs and scenarios. To the best of our knowledge, this model is the first to incorporate multiple media sources and serves as a benchmark to evaluate whether VCAs achieve optimal QoE in multimedia transmissions.

Navigating our research, we encounter several vital issues. First, acquiring QoE metrics like data rate, resolution, and latency from closed-source commercial VCAs is difficult. Second, to effectively gain real users' preferences from a user study, we need to design media source combinations that reflect a variety of network conditions. It is challenging to select a representative subset of these combinations for our user study while ensuring that we do not sacrifice the thoroughness and scope of our research. Third, building a general and robust QoE prediction model that applies to all scenarios and VCAs is essential.

**Measurement of VCAs:** To extract QoE metrics, we devise a measurement methodology to collect data from three VCAs. For large-scale controlled laboratory experiments, we engineered an automation tool responsible for client emulation, network control, and data aggregation at the client end. Among more than 20 hours of video sessions, we discovered their bandwidth allocation strategies and identified commonalities. Further, to explore the characteristics of individual media source transmission, we conduct an extensive

case study on Zoom under restricted network conditions, specifically focusing on scenarios where bandwidth is limited and packet loss is high. Our significant findings are presented as follows:

- Under four scenarios with different combinations of media sources, the three VCAs consistently prioritize bandwidth allocation in the same order: Audio > Screen > Video.

- Zoom applies distinct bitrate adaptation strategies for video and screen. Specifically, it supports three-resolution video transmission and one-resolution screen transmission. However, this fixed strategy may not continuously satisfy user expectations under different scenarios.

**User Study:** Our IRB-approved user study successfully gathered 45,000 user ratings from 800 participants via Amazon Mechanical Turk [20] and covers four common usage scenarios. We formulate bitrate combination samples by merging different quality levels of three media sources.

Evaluating the QoE over such a wide range of bitrate combinations is challenging, primarily because comparing every possible combination with one another in a user study is impractical. To overcome this, we introduce an "accumulated score" method that allows us to compare two consecutive combinations as an alternative to comparing each possible pair. As a result, we can conduct a user study with only a fraction of the total combinations and still gain insight into user preferences across all possible pairs.

**QoE Modeling:** To interpret user ratings and define preference relationships, we employ the PageRank algorithm [2]. We then rank the combinations of media source bitrates based on the PageRank scores. Combinations that receive higher scores are identified as more preferred by users, establishing a clear preference hierarchy.

Following this, we develop a QoE prediction model that can be generalized to evaluate QoE across various VCAs and scenarios. This model is capable of predicting the QoE values for any given set of input combinations, enabling us to determine if an input combination achieves optimal QoE. Additionally, it allows us to rank a set of combinations, pinpointing which one offers the best QoE. This capability provides significant insights and actionable recommendations, guiding VCAs to improve user experience by fine-tuning their services to meet optimal user preferences, particularly in bandwidth-constrained environments.

Applying this model to evaluate Zoom, Webex, and Google Meet, we find that their performance is far away from the optimal QoE as predicted by our model. Among them, Zoom stands out by always offering a better QoE, showcasing its superior ability to manage bandwidth and adapt to varying network conditions. Nonetheless, all platforms have room for improvement to reach the optimal QoE.

The contributions of this paper can be summarized as:

- **Key observations and takeaways of VCAs:** We perform measurements of bandwidth allocation on three VCAs: Zoom, Webex, and Google Meet, providing valuable insights into their designs.
- **QoE Modeling:** We introduce a pioneering QoE prediction model that uniquely incorporates multiple media sources and is adaptable to a variety of scenarios across different VCAs.
- **Dataset:** Our Dataset, comprising 20 hours of video conferencing sessions and feedback from 800 study participants, offers valuable resources for future research. The dataset will be opensourced if our paper is accepted.

This research does not raise any ethical issues.

## 2 RELATED WORK

**Measurement of Video Conferencing:** Different Video Conferencing Applications (VCAs) use the same communication protocols but differ in their choice of codecs and traffic control strategies. This results in varied performance, even under identical network conditions. Macmillan et al. [16] measured Zoom, Google Meet, and Microsoft Teams, revealing distinctions in their recovery methods, video quality adaptation, and network utilization. [5] highlights comparative results of streaming lag, audio/video QoE, and resource consumption among Zoom, Webex, and Google Meet. [29] evaluates system architecture, resilience to loss, and audio/video QoE for Google+, iChat, and Skype. [21] evaluate the performance of WebRTC-based Video Conferencing, including processing delay, CPU utilization, latency, jitter, packet Loss, and packet delay.

QoE measurements are paramount when assessing VCAs. In terms of audio, commonly evaluated metrics include audio quality [16] and audio latency [29]. Video QoE assessments encompass aspects like framerate [15, 16], resolution [15], latency [29], and overall video quality [16]. Beyond these network-level analyses, researchers also delved deeper, employing transport-layer analysis to uncover the inner designs, such as congestion control [4, 12, 22], mechanisms for packet loss recovery [29], measurement-driven functional model [12], etc.

Some studies explore the security issues of VCAs. [8] conduct a dynamic security analysis of Zoom, Google Meet, and Microsoft Teams. [14] investigates three versions (desktop, web, smartphone) of WebEx and identifies several relevant artifacts, including user account information, encryption keys, media/text files, meeting records, etc. [18, 28] scrutinize Zoom's encryption method, offering insights and methodologies for decoding UDP and RTP packets.

**QoE Modeling:** [1] developed a predictive model of QoE for internet video. [7] conducted a small-scale user study to develop a QoE model for evaluating real-time video systems. [19] developed a QoE model to map network QoS metrics to video streaming QoE. [30] conducted a user study to model the QoE of 360-degree volumetric video streaming. However, these existing studies only focus on video sources without considering audio and screen-sharing media sources or their combined QoE.

## 3 MEASUREMENT OF VCAS

In this section, we conduct a thorough analysis of three VCAs: Zoom, Webex, and Google Meet. Our focus centers on exploring their bandwidth allocation strategies for different media sources, including video, audio, and screen.

### 3.1 Measurement Methodology

**An Automation Tool.** To effectively control VCAs and simulate human activities programmatically, we develop a command-line automation tool, enabling efficient client emulation, network control, and data collection. It facilitates a streamlined process for conducting our experiments, as depicted in Figure 1.

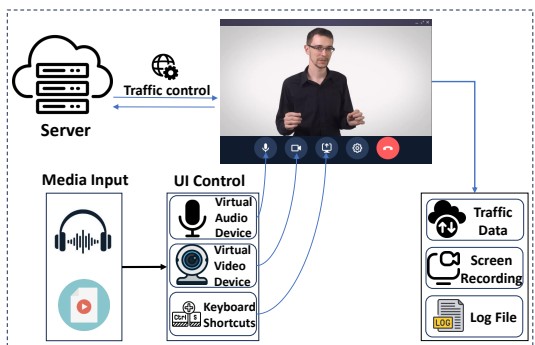

**Figure 1: Testbed for measuring commercial VCAs.**

• *Client Emulation.* To facilitate the automated sending and receiving of media sources, this tool incorporates *snd-aloop* modules and *aplay* [9] for audio input playback, along with *v4l2loopback* [10] modules paired with *FFmpeg* [26] for video input playback. We also utilize xdotool [25] to programmatically execute keyboard and mouse commands for various VCA operations, such as starting/ending screen, enabling/disabling audio/video, switching view layout, and opening/closing full-screen mode.

• *Network Control.* For managing network conditions on the client side, we use Linux Linux *TC* [3], allowing us to configure uplink and downlink bandwidth and adjust latency precisely.

• *Data Collection.* For our analysis of bandwidth allocation across three VCAs, we capture network traffic via *tcpdump* [11] and obtain QoE metrics of each media source.

To collect QoE metrics, we set up the video or screen sharing to display in full-screen mode and anchor the statistics panel at the bottom-left corner of the screen. This panel continuously displays real-time data on resolution and frame rate, as illustrated in Figure 1. For VCAs that offer detailed log files, we download these logs periodically. Then, we can extract frame rate and resolution data, considering their averages as session metrics. QR code recognition helps synchronize sender and receiver frames, facilitating SSIM and FPS calculations.

For our detailed case study on Zoom, we gather extensive information from the decoded UDP and RTP packets. We refer to methods in [17, 18] to analyze UDP and RTP packet headers, identifying valuable details about the media sources' transmission.

**Experimental Setup.** Our measurement framework operates on machines running Ubuntu 22.04.1 LTS with Zoom 5.17.11, Webex 43.2, and Google Meet installed. These machines are connected to our on-campus wireless network, which guarantees a minimum bandwidth of 90 Mbps for both uploads and downloads. Each experiment includes $N (N \geq 2)$ users, where one user (referred to as the "sender") is responsible solely for uploading media to the VCA servers. Our research encompasses three measurements, each grounded in its unique experimental setup.

• *Bandwidth Allocation for three VCAs:* For our study on bandwidth allocation across three VCAs—Zoom, Webex, and Google Meet, we identify four key scenarios reflecting different VCA configurations and user behaviors, detailed in Table 1. These scenarios are composed of varied combinations of media sources and window sizes. Our experiments are unidirectional, focusing on these

scenarios to evaluate the data rate of each media source under two main conditions: a) a limited uplink bandwidth at the sender's end and b) a constrained downlink bandwidth at the receiver's end. For both conditions, we set bandwidth limits at intervals of 0.2,0.4,0.6,0.8,1.0 Mbps. We schedule each video conferencing session to last five minutes and conduct each experiment three times to ensure reliability.

• *Zoom Measurement:* To examine the transmission behavior of each media source under network constraints, our experiments center on Zoom and involve one-directional tests with $N = 6$ participants. These tests separately address bandwidth and packet loss restrictions to understand their distinct impacts. **Bandwidth:** The setup includes one sender with unrestricted bandwidth, while the five receivers have unique downlink capacities, specifically {None (no limits), 750, 500, 250, 150} kbps. **Packet loss:** In this setup, one sender operates with an unrestricted network, whereas the five receivers experience 10%, 20%, 30%, 40%, 50% packet loss. **Media Inputs.** For our audio input, we use a recording of a lecture where the lecturer engages in continuous speech, ensuring a consistent audio profile for the duration of our study. In terms of video and screen inputs, we select a lecture video, which is standardized to a resolution of 1280x720 and runs at a frame rate of 25 FPS. To facilitate precise alignment of transmitted frames with their received counterparts, we embed a QR code for each frame of the video content. This methodological detail enhances the reliability of our frame-by-frame analysis.

## 3.2 Bandwidth Allocation across Media Sources

In practical video conferencing sessions, simultaneous use of multiple media sources is the norm. This section explores how VCA prioritizes and distributes bandwidth when multiple media sources are in play, especially under bandwidth-constrained conditions. We present four frequently encountered scenarios with distinct combinations of media sources, as outlined in Table 1.

|  | Audio | Video | Screen |
|---|---|---|---|
| **Scenario 1** | √ | (Full-Screen) | |
| **Scenario 2** | √ | | (Full-Screen) |
| **Scenario 3** | √ | (Thumbnail) | (Full-Screen) |
| **Scenario 4** | √ | (Half-Screen) | (Half-Screen) |

**Table 1: Four common scenarios with different media source inputs.**

**Scenario 1 (Figure 2(a)):** One example of this scenario is the online interview; body language and facial expressions play a pivotal role. As bandwidth becomes limited, the video data rate declines while the audio data rate remains consistent (around 100Kbps). Interestingly, at extremely low bandwidths, around 200Kbps, Zoom prioritizes audio quality, elevating its data rate and causing the video transmission to diminish almost entirely.

**Scenario 2 (Figure 2(b)):** This scenario can be applied to group discussions, such as academic deliberations, where the focus is on slides or whiteboard content. Though Scenario 2 displays a similar trend to Scenario 1, it diverges because the audio data rate increases when bandwidth narrows to 400Kbps. Notably, even with severely constrained bandwidth, the screen transmits at a data rate, albeit lower than audio.

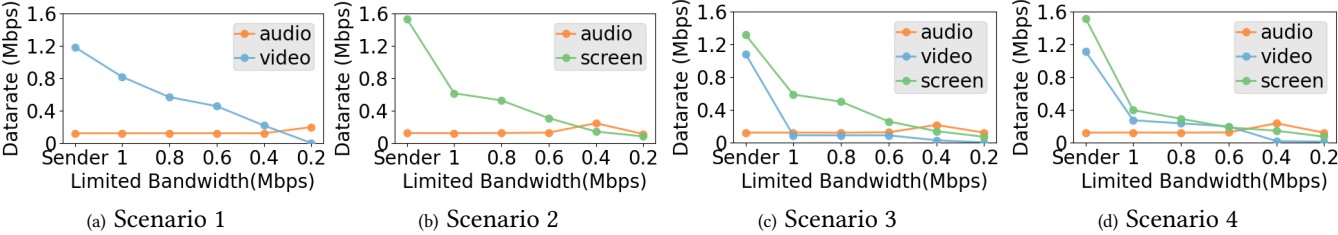

(a) Scenario 1     (b) Scenario 2     (c) Scenario 3     (d) Scenario 4

Figure 2: Zoom: average bandwidth under four scenarios with downlink bandwidth limits on each receiver

**Scenario 3 (Figure 2(c)):** This scenario commonly happens in an online conferencing where a lecturer presents their work, and attendees primarily listen. A noticeable data rate discrepancy exists between the Sender and 1M for video. This is attributed to Zoom displaying the video in a minimized thumbnail window, thus allocating minimal bandwidth to it. As bandwidth constraints tighten, audio and video data rates stabilize at approximately 120kbps and 85Kbps, respectively. As the available bandwidth further drops to 400Kbps, the audio data rate increases at the expense of other sources. At an extreme bandwidth constraint of 200kbps, the video data rate becomes negligible, whereas others decrease, with audio maintaining a higher rate than the screen.

**Scenario 4 (Figure 2(c)):** The typical usage example of this scenario is Big Tech companies' product launch events, where slides provide detailed information. At the same time, the presenter visually demonstrates the product's features and functionalities in real-time. When the bandwidth is restricted, both the video and screen data rate drops correspondingly while the audio data rate remains unchanged. At the extremely poor bandwidth, Scenario 4 displays a similar trend to Scenario 3.

The same conclusions are applicable when considering scenarios with limited uplink bandwidth, as detailed in the supplementary material. Moreover, Zoom's conclusions regarding scenarios also apply to Webex and Google Meet, as detailed in the supplementary material. While these platforms exhibit varied data rates for each media source under different network bandwidth conditions, their bandwidth allocation prioritization remains consistent.

> **Findings:** Although three VCAs implement distinct bandwidth allocation strategies, they have the same bandwidth allocation prioritization: audio >screen >video.
> **Takeaways:** This fixed traffic prioritization for audio, video, and screen may degrade the user experience, as it may not match users' varying demands for these media sources based on their different meeting purposes.

## 3.3 Case Study on Zoom

To gain insights into the transmission of individual media sources, we carry out a focused case study on Zoom. This case study examines Zoom's adaptive bitrate strategies in constrained networks, specifically those with limited bandwidth and increased latency.

### 3.3.1 Impact of Bandwidth Limits

● *Audio Transmission*

For audio-only conferencing, we observe that a consistent average bitrate of 120kbps is maintained. We do not further apply

bandwidth restrictions on audio transmission because we discover that a bandwidth lower than 150kbps jeopardizes the stability of the meeting connection.

● *Video Transmission*

In multi-user video conferencing, when the downlink bandwidth declines at receivers, the data rate decreases correspondingly, resulting in the varying degradation of QoE metrics. This degradation mainly first affects the framerate and then the resolution. As shown in Table 2, Receiver1, Receiver2, and Receiver3 have the same resolution but a descending framerate along with the decline of data rate. Then, the degradation happens on resolution. With the continuous decline in data rate, the resolution decreases to 320×180 (180p) in Receiver4 and 256 × 144 (144p) in Receiver5. Accordingly, we discover the obvious decline in SSIM value. However, the framerate of Receiver4 and Receiver5 does not drop too much. The detailed packet-level analysis via UDP/RTP decoding can be found in the supplement material.

| | Sender (None) | Receiver1 (None) | Receiver2 (750k) | Receiver3 (500k) | Receiver4 (250k) | Receiver5 (150k) |
|---|---|---|---|---|---|---|
| Data rate(kbps) | 1158±120 | 883±130 | 647±85 | 453±44 | 218±35 | 144±20 |
| Framerate(FPS) | 21±3 | 21±3 | 13±2 | 10±2 | 8±1 | 7±2 |
| Resolution | 360p | 360p | 360p | 360p | 180p | 144p |
| SSIM | | 89±3 | 86±2 | 85 ±1 | 80 | 75 ±1 |

**Table 2: QoE metrics of Video with Bandwidth Limits**

● *Screen Transmission*

Unlike video, the screen maintains a consistent resolution. Regardless of the downlink bandwidths allocated to each receiver during a session, the resolution remains consistent across sender and receivers. In fact, the resolution at the receiver's end mirrors that of the sender's screen-sharing content. If there's a change in the sender's resolution, the receivers adjust accordingly. As evidenced in Table 3, a decline in downlink bandwidth affects corresponding drops in data rate and framerate. Intriguingly, even when the framerate nears zero, the resolution remains unchanged across all receivers, and there's only a little dip in SSIM. This suggests that degradation in screen quality predominantly impacts the framerate.

| | Sender (None) | Receiver1 (None) | Receiver2 (750k) | Receiver3 (500k) | Receiver4 (250k) | Receiver5 (150k) |
|---|---|---|---|---|---|---|
| Data rate(kbps) | 1482±230 | 1439±230 | 547±150 | 326±85 | 168±40 | 118±20 |
| Framerate(FPS) | 10±2 | 10±2 | 4±1 | 2±1 | 1±1 | <1 |
| Resolution | 720p | 720p | 720p | 720p | 720p | 720p |
| SSIM | | 89±2 | 87±3 | 85±3 | 83 ±2 | 82 ±1 |

**Table 3: QoE metrics of Screen with Bandwidth Limits**

**Findings:** Zoom employs distinct bitrate strategies for video and screen. Videos prioritize framerate, sacrificing resolution in low-bandwidth situations, whereas the screen opts for higher resolution at the expense of framerate.

**Takeaways:** The one-resolution screen-sharing transmission and three-resolution video transmission don't adapt to various factors such as network conditions (*e.g.,* available bandwidth) and user configurations (*e.g.,* window size of the screen-sharing content), incurring network resource waste and QoE degradation. Thus, Zoom can strategically offload a part of the transcoding workload to more powerful Zoom servers, which also reduces uplink bandwidth usage, or find more intelligent adaptation strategies to balance the trade-off between the additional transcoding overhead and the quality/latency requirement.

### 3.3.2 Impact of Packet Loss

• *Audio Transmission*

When packet loss increases, the bitrate of audio transmission remains constant rather than decreasing proportionally. To preserve stable audio connections, retransmission of lost packets is initiated. This approach ensures that audio quality is maintained despite network constraints, emphasizing Zoom's priority to delivering uninterrupted and clear audio communication.

• *Video Transmission*

| | Sender (None) | Receiver1 (10%) | Receiver2 (20%) | Receiver3 (30%) | Receiver4 (40%) | Receiver5 (50%) |
|---|---|---|---|---|---|---|
| Data rate(kbps) | 1002±185 | 901±85 | 724±90 | 621±45 | 565±25 | 387±32 |
| Framerate(FPS) | 24±1 | 22±1 | 20±1 | 16±2 | 12±2 | 8±2 |
| Resolution | 360p | 360p | 360p | 360p | 360p | 360p |
| SSIM | | 89±1 | 88±2 | 86 ±2 | 87 | 85 ±1 |

**Table 4: QoE metrics of Video with Packet Loss**

In multi-user video sessions, an increase in packet loss at the receiver ends leads to a corresponding decrease in the data rate. This reduction primarily impacts the video's framerate while the resolution remains unchanged. As illustrated in Table 4, despite the five receivers maintaining the same video resolution, there is a notable decrease in framerate in conjunction with the declining data rate. Accordingly, the SSIM values experience slight degradation.

• *Screen Transmission*

| | Sender (None) | Receiver1 (10%) | Receiver2 (20%) | Receiver3 (30%) | Receiver4 (40%) | Receiver5 (50%) |
|---|---|---|---|---|---|---|
| Data rate(kbps) | 1280±230 | 854±88 | 303±90 | 199±65 | 123±30 | 85±25 |
| Framerate(FPS) | 15±3 | 12±2 | 7±4 | 3±2 | 1±1 | ≤±1 |
| Resolution | 720p | 720p | 720p | 720p | 720p | 720p |
| SSIM | | 91±1 | 90±2 | 88 ±2 | 88 ±1 | 88 ±1 |

**Table 5: QoE metrics of Screen with Packet Loss**

Similar to video transmission, the screen maintains a consistent resolution and declining framerate. As evidenced in Table 5, an increase in packet loss precipitates corresponding drops in data rate and framerate. This suggests that degradation in screen quality predominantly impacts the framerate.

| | | |
|---|---|---|
| Age | 18-25: 25.8%, 26-30: 27.0% | |
| | 31-35: 16.4%, 35+: 30.8% | |
| Gender | Male: 60.3%, Female: 39.2% | |
| | Other: 0.5% | |
| htbp | US: 50.0%, IN: 30.1%, | |
| Country | BR: 4.0%, IT: 5.7%, | |
| (30 Total) | UK: 2.2%, Other: 6.1% | |
| Education | Bachelor: 50.1%, Master: 26.3% | |
| | Ph.D.:8.1%, Other: 15.5% | |

**Table 6: Demographics of the 800 subjects in our user studies.**

**Findings:** When experiencing packet loss, Zoom employs a retransmission mechanism to ensure the clarity of audio connections. In contrast, Zoom opts to reduce the framerate rather than the resolution for video and screen, prioritizing clarity over fluidity and aiming to preserve essential details even under challenging network conditions.

**Takeaways:** Zoom's strategies for managing packet loss prioritize clarity in audio and visuals, but this doesn't always meet user expectations. Users often seek a balance between clarity and fluidity, preferring not to experience significant lags for the sake of sharpness. Thus, a balance between clarity and fluidity is essential to satisfy user needs better and enhance the overall conferencing experience.

## 4 USER STUDY

While VCAs have provided insights into their bandwidth allocation strategies under constrained network conditions, it remains unclear if these strategies align with user preferences or yield the optimal user experience. To bridge this gap, we begin with an IRB-approved user study to collect a dataset of real users' preferences on bandwidth allocation for diverse media sources in VCAs under constrained networks.

### 4.1 Methodology

Our user study methodology follows the double stimulus comparison scale (DSCS) method recommended by ITU (International Telecommunication Union) [23]. In this approach, participants watch the same 15-second video conferencing clip twice back-to-back, each viewing featuring a different media source bitrate combination. Afterward, they subjectively compare their perceived Quality of Experience (QoE) using a seven-choice scale ("The first one is {much better, better, slightly better, similar to, slightly worse, worse, much worse} than the second one."). For data processing purposes, these qualitative choices are converted into numerical values, with the scale translating to numbers from 1 to 7. To prevent audio interference between two video clips, participants are instructed to manually click the "play" button to view each clip sequentially. This strategy offers two significant benefits: (1) making it easier to scale up the user study and (2) engaging a globally diverse pool of participants.

Instead of conducting an in-person user study, we opt for an online approach using Qualtrics [27] and Amazon Mechanical Turk (AMT) [20].

**Dataset Overview.** To ensure the broad applicability of our findings, our user study meticulously covers the four representative

video conferencing scenarios outlined in Table 1. For each scenario, we craft distinct content for different media sources. The study engages 800 participants, with their demographics detailed in Table 6. Specifically, Scenarios 1 and 2 involve 100 participants each, while Scenario 3 and Scenario 4 have 300 participants, respectively. Collectively, this approach yields a dataset comprising over 45,000 user ratings.

## 4.2 Generate Bitrate Combination Samples

Given a specific bandwidth $B$, the potential bitrate combinations for distributing it among various media sources are infinite. Rather than attempting to enumerate an exhaustive list of these combinations, we need to strategically select a finite and representative set of bitrate combination samples. Inspired by Zoom's bitrate adaptation strategy, we create several quality levels for each media source. These differentiated quality levels of media sources are then combined, forming a carefully selected set of bitrate combination samples.

We begin by producing benchmark media sources: an audio stream at 128kbps, a video at 720p resolution with 25FPS and a bandwidth of 1.5Mbps, and a screen feed also at 720p and 25FPS consuming 1.5Mbps. Then, we transcode these benchmarks across a spectrum of quality levels. As shown in Table 7, we create 3 levels for audio and 9 levels (2 FPS levels X 3 resolution levels) for both video and screen. By combining different media sources together, we craft a set of 27(3x9), 27(3x9), 243(3x9x9), and 243(3x9x9) bitrate combination samples for Scenario 1, 2, 3, and 4, respectively.

| audio | 128kbps | 32kbps | 8kbps |
|---|---|---|---|
| video | 25FPS | 15FPS | 5FPS |
| | 720p | 360p | 180p |
| screen | 25FPS | 15FPS | 5FPS |
| | 720p | 360p | 180p |

Table 7: Different quality levels of audio, video, and screen

## 4.3 Calculate User Rating

$$U = \begin{bmatrix} 0 & u_{1,2} & \cdots & \cdots & u_{1,n} \\ \vdots & 0 & u_{2,3} & \cdots & u_{2,n} \\ \vdots & 0 & \ddots & \ddots & \vdots \\ \vdots & 0 & \cdots & 0 & u_{n-1,n} \\ 0 & \cdots & \cdots & \cdots & 0 \end{bmatrix} \quad (1)$$

To evaluate QoE across numerous bitrate combinations, we face the challenge of managing the vast number of pairwise comparisons. With $N$ bitrate combination samples, the exhaustive pairwise comparison approach would necessitate N(N-1)/2 comparisons, which becomes unfeasible as $N$ increases. Specifically, we need 2351 comparisons in Scenarios 1 and 2 and 29403 in Scenarios 3 and 4, which is clearly impractical. To address this, we propose the "accumulated score" method. It allows us to conduct $N$ comparisons but still receives results that closely approximate results from $N(N-1)/2$ comparisons [13]. Essentially, this method enables us to deduce the entire user rating matrix $U$ (as shown in Matrix 1) by examining only a fraction of its elements. Here's how it works:

**Rank Combination Samples:** We rank all $N$ combination samples by their bitrate, based on the assumption that a higher bitrate typically means higher user preference. We compare each pair of adjacent combinations, $N_i$ and $N_{i+1}$, where $i$ ranges from 1 to $N-1$. This yielded $N-1$ user ratings, namely $u_{i,i+1}$.

**Calculate Accumulated Score:** After comparison, we will get $N$ use ratings ($u_{i,i+1}$). We set the accumulated score for the combination with the lowest bitrate (the Nth combination) to 0, namely $u_{N-1,N} = 0$. The accumulated score for the $N-1$th combination is calculated by adding the accumulated score of the $N$th combination ($acc\_score_N$) with the user rating obtained from the $N-1$th and $N$th combination comparison ($u_{N-1,N}$). Accordingly, we apply this calculation sequentially to determine the accumulated scores for all $N$ combinations by using the formula 2. These $N$ accumulated scores are calculated on a per-user basis.

$$\begin{aligned} acc\_score_N &= u_{N-1,N} = 0 \\ acc\_score_i &= acc\_score_{i+1} + u_{i,i+1}, \quad i \in \{1, N-1\} \end{aligned} \quad (2)$$

**Obtain All user ratings.** After obtaining the accumulated scores for $N$ combinations, we are able to determine the user rating between any two combinations by calculating the difference in their accumulated scores, as shown in Formula 3. This approach enables us to populate all the necessary elements in Matrix 1.

$$u_{i,j} = acc\_score_i - acc\_score_j, \quad i, j \in \{1, N-1\} \quad (3)$$

## 5 QOE MODELING

To understand user ratings and user preference relationships, we employ the PageRank algorithm [2] to establish a clear preference hierarchy and derive QoE values. Building on these insights, we create a QoE prediction model to predict QoE values for any given media source combinations.

## 5.1 QoE values

The PageRank algorithm evaluates our user study results using a directed graph. Each node within this graph symbolizes a distinct bitrate combination, with edges between nodes representing comparative user ratings that highlight preference relationships. Here's a more detailed breakdown of the process:

• Node Creation: Each node in the graph corresponds to a unique media source bitrate combination. These combinations are directly derived from the scenarios presented in our user study.

• Edge Construction and Weight Assignment: The graph's edges are established based on the user ratings collected during the study. Participants are given seven options to express their preference between two combinations, ranging from "Combination A is much better, better, slightly better, similar to, slightly worse, worse, much worse than Combination B". These verbal options are then converted into a numerical scale that ranges from 3 to -3, reflecting the degree of preference. An edge is drawn from node B to node A if the rating indicates a preference for Combination A (rating > 0). Conversely, if the preference leans towards Combination B (rating < 0), an edge is drawn from node A to node B. The magnitude of the rating is used to assign weight to each edge, quantitatively expressing the degree of preference.

• Assign QoE Values: PageRank calculates scores for each node, effectively indicating the level of user preference for each bitrate combination. We then rank the media source bitrate combinations, identifying those with higher scores as more favored by users. Following this ranking, we assign QoE values based on each combination's position in the preference hierarchy; the top-ranked or most preferred combination receives the highest possible QoE value, while the least favored combination is assigned a QoE value of 1. For scenarios 1 and 2, which feature 27 combinations, the QoE value for the highest-ranked combination is 27. Similarly, scenarios 3 and 4, each with 243 combinations, see their most preferred combination receiving a QoE value of 243.

## 5.2 Model Design

The input parameters for the QoE model, specifically designed to accommodate different scenarios, are detailed in Table 8.

| category | parameter |
|---|---|
| **audio** | [audio bitrate] |
| **video** | [video resolution, video framerate] |
| **screen** | [screen resolution, screen framerate] |
| **bandwidth** | [overall bitrate] |
| **others** | [the ratio of window size between video and screen] |

Table 8: Input parameters of each media source

• Scenario 1: The input vector includes parameters specific to audio and video, along with the total bitrate.

• Scenario 2: This vector is associated with audio, screen-sharing, and the overall bitrate.

• Scenario 3 and 4: The input vector is all-encompassing, drawing parameters from every category, notably audio, video, screen-sharing, and the total bandwidth.

• General: The broad input vector aggregates parameters from all relevant categories—audio, video, screen-sharing, total bandwidth—and incorporates the newly introduced parameter of the window size ratio between video and screen-sharing. This approach ensures our QoE model's generality and relevance across different video conferencing scenarios.

In our analysis, we explore four distinct models, each meticulously adjusted to optimize our prediction task:

(1) Logistic Regression: we set $tol = 10e - 6$, $random\_state = 0$, and $solver =$"$newton - cholesky$"; This model is configured with a tolerance level (tol) of $10^{-6}$, a $random\_state$ set to 0 for reproducibility, and utilizes the $newton - cholesky$ method as its solver.

(2) Random Forest Regression (RF): For the Random Forest model, we specify $n\_estimators = 100$, indicating the number of trees in the forest, and maintain a $random\_state$ of 0 to ensure consistent results across different runs.

(3) Gradient Boosting Decision This model employs $n\_estimators =$ 10, reflecting a more conservative approach with ten trees, and a $learning\_rate$ of 0.1, balancing the speed and accuracy of learning.

(4) Multi-layer Perceptron Regression (MLP): The MLP model is adjusted with a learning rate (alpha) of $10^{-6}$, employs a "$logistic$" activation function, an "$adaptive$" learning rate to adjust as learning progresses, a $random\_state$ of 0 for reproducibility, and $max\_iter =$ 2000, allowing a generous number of iterations for convergence.

## 6 EVALUATION

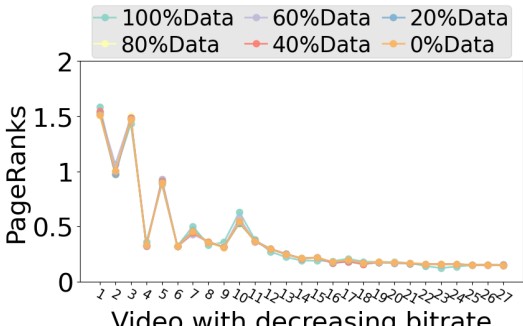

Figure 3: PageRank results with missing data

### 6.1 Efficiency of Accumulated Score

To enhance the cost-effectiveness of our user study, we adopt a novel strategy known as the "accumulated score." This method focuses on evaluating consecutive combinations of media source bitrates as an alternative to comparing every possible pair. We validate this approach through a simulation where five participants thoroughly assess each potential pair of combinations for Scenario 1, enabling us to directly create user rating matrices $U$.

For validation, we sequentially utilized 0%, 20%, 40%, 60%,80% and 100% of the user rating data in $U$, interpolating missing values using the accumulated score technique. This process resulted in six matrices: $U_0$, $U_{20}$, ..., $U_{100}$. $U_0$ represents the application of our method. Then, we compute the PageRank for every combination sample across these six matrices, whose results are depicted in Figure 3.

To assess the consistency of our method ($U_0$) with the approach that involves comparing every pair ($U_{100}$), we used the Sequence-Matcher to calculate the similarity in PageRank ranking between them. The similarity trends illustrated in Figure 3 and an average similarity score of 0.88 from the SequenceMatcher strongly affirm the effectiveness and reliability of our accumulated score methodology.

### 6.2 QoE Modeling Evaluation

Our Quality of Experience (QoE) model is adept at predicting QoE values for specific combinations of media sources. This capability allows us to determine whether a given combination achieves the optimal QoE. Furthermore, when presented with multiple combinations, the model enables us to rank them based on their QoE performance. To assess the reliability and accuracy of these predictions and rankings, we employ 10-fold cross-validation [31], a robust statistical technique that ensures each data point is used for both training and testing across the validation process. This method provides a comprehensive gauge of the model's performance, ensuring its predictions are both precise and reflective of real-world user experiences.

• **QoE Prediction Evaluation:** We evaluate the accuracy of our QoE predictions using two key metrics: Root Mean Squared Error (RMSE) and Mean Absolute Error (MAE). Lower values in these metrics indicate more accurate predictions, closely matching the actual QoE values from user studies. According to Table table 9,

| Scenario | Logistic | | | RF | | | GBDT | | | MLP | | |
|---|---|---|---|---|---|---|---|---|---|---|---|---|
| | *MAE* | *RMSE* | *Accuracy* | *MAE* | *RMSE* | *Accuracy* | *MAE* | *RMSE* | *Accuracy* | *MAE* | *RMSE* | *Accuracy* |
| Scenario 1 | 0.12 | 0.92 | 82.13% | 0.12 | 0.86 | 81.3% | 0.15 | 0.93 | 82.1% | 0.12 | 0.83 | 84.61% |
| Scenario 2 | 0.12 | 0.96 | 81.90% | 0.13 | 0.87 | 82.15% | 0.16 | 0.97 | 82.20% | 0.11 | 0.85 | 84.55% |
| Scenario 3 | 0.15 | 2.81 | 78.79% | 0.13 | 2.50 | 81.12% | 0.16 | 2.87 | 81.04% | 0.13 | 2.19 | 84.37% |
| Scenario 4 | 0.14 | 2.92 | 80.48% | 0.11 | 2.28 | 82.06% | 0.15 | 2.67 | 81.49% | 0.12 | 2.24 | 82.62% |
| General | 0.19 | 4.06 | 70.51% | 0.09 | 1.75 | 81.63% | 0.13 | 2.56 | 81.79% | 0.08 | 1.78 | 82.86% |

**Table 9: Comparisons of the average MAE, RMSE, Accuracy (%) with Logistic, Random Forest, GBDT, MLP algorithms. The best results are shown with underline**

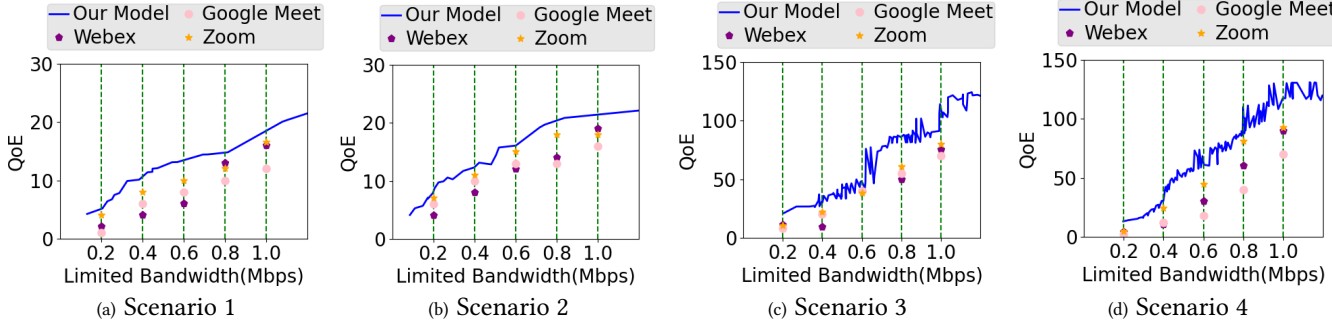

(a) Scenario 1  (b) Scenario 2  (c) Scenario 3  (d) Scenario 4

**Figure 4: QoE comparison between Zoom, Webex, and Google Meet. Figures only show scenarios under 1.2Mbps**

all tested scenarios show competitive MAE and RMSE scores, with Scenarios 1 and 2 demonstrating slightly better performance, likely due to the less complex nature of their bitrate combinations. Notably, the Multi-layer Perceptron (MLP) model achieves the lowest RMSE and MAE across scenarios.

• **Combination Sets Ranking Evaluation:** We assess the model's accuracy in ranking various combinations by comparing its predicted QoE rankings to those derived from actual user feedback. According to Table table 9, the models show promising performance overall, with accuracy exceeding 70%. The MLP model, in particular, distinguishes itself by consistently achieving accuracy rates above 80% in all scenarios.

In conclusion, our evaluation metrics underscore the model's effectiveness in precisely predicting QoE and in accurately ranking combinations. The MLP model stands out as particularly adept, surpassing other models in every scenario tested. The robust performance across various diverse scenarios highlights its generality and suitability for enhancing user experience within VCAs.

## 6.3 QoE Evaluation of Three VCAs

In §3.2, we investigate the bandwidth allocation of three VCAs. While this measurement provided insights into them, it leaves open the question of whether these strategies truly align with user preferences or achieve the best possible user experience.

To address this, we apply our general QoE model to predict Zoom, Webex, and Google Meet's QoE under restricted downlink bandwidth conditions (0.2, 0.4, 0.6, 0.8, 1) Mbps. We treat the QoE results from our user study as a benchmark (optimal QoE), against which we compare VCA's actual QoE performance.

As depicted in Figure 4, under the same bandwidth constraints, the performance of the three Video Conferencing Applications (VCAs) — Zoom, Webex, and Google Meet is far away from the optimal QoE. Among these, Zoom demonstrates a higher QoE value relative to Webex and Google Meet in most scenarios, suggesting its bandwidth allocation strategies are more effective. Notably, the contrast between our benchmark and predicted QoE values from the VCAs becomes more pronounced in Scenarios 2, 3, and 4 compared to Scenario 1. This significant difference is likely influenced by screen-sharing, which appears to affect the QoE outcome more severely in these scenarios.

## 7 CONCLUSION

Our research delves into the multimedia transmission capabilities of Video VCAs, with a particular focus on three key media sources: audio, video, and screen. Initially, we examine the bandwidth allocation strategies of three prominent VCAs—Zoom, Webex, and Google Meet—paying special attention to their performance in networks with limited bandwidth. Following this, we present a detailed case study on Zoom to explore its bitrate adaptation strategies for each media source when faced with network constraints about bandwidth limits and packet loss. Building on these analyses, we propose a QoE model designed to predict QoE performance across various scenarios and platforms accurately. The findings from our evaluation demonstrate the model's effectiveness and generality. This model serves as a tool for VCAs to improve user experience by providing valuable insights and recommendations, particularly in scenarios with limited network resources.

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
