# OpenReview forum: "An In-depth Study of Bandwidth Allocation across  Media Sources in Video Conferencing"
_acmmm.org/ACMMM/2024/Conference — MM2024 Oral_

### Official Review · Reviewer_G3ME · 2024-05-04

**Rating:** 5
**Confidence:** 3

**Summary:**

The paper discusses how popular Video Conferencing Applications (VCAs) allocate bandwidth among various media sources, namely audio, video, and screen-sharing. It delves into the design of an automation tool to reveal the bandwidth allocation strategies of Zoom, Webex, and Google Meet across four scenarios, defined by the presence and window size of the three media sources. Zoom is then singled out for a detailed examination of its adaptation strategies for each media source under degraded network conditions. To construct a Quality of Experience (QoE) model for VCAs, a subjective assessment involving over 800 participants was conducted via a crowdsourcing platform, and the resulting dataset will be made publicly available. This assessment utilized the "accumulated score" method, while the modeling phase employed the PageRank algorithm and Multi-Layer Perceptron (MLP). Subsequently, this model was employed to evaluate the QoE provided by the adaptation strategies of the three VCAs, revealing discrepancies from optimal performance.

**Strengths:**

• Although not explicitly claimed as a contribution, the automation tool itself is a significant contribution.

• The paper is well-written, clear, and addresses a pertinent and timely topic.

• The results are intriguing and hold interest.

• The crowdsourcing effort involving 800 participants is commendable.

• The utilization of techniques such as the “accumulated score” method and “PageRank” appears apt and clever, with their effectiveness being convincingly demonstrated.

• The analysis and insights provided are persuasive and pioneering within the field.

**Limitations:**

• The focus solely on Zoom may appear limited, likely due to its suitability for the techniques mentioned (cited as 17-18 for packet analysis). Exploring other platforms would have added significant value.

• There is a lack of consideration for congestion control algorithms or their parameters in experiments involving multiple receivers.

• The extreme degradation of bandwidth in both experiments, across all three VCAs and specifically Zoom, seems excessively severe, bordering on unrealistic.

• The resolution of the video transmitted by Zoom (360p) may be excessively low, potentially justifying the limited resolution set of the receivers.

• Minor typos are present throughout the paper, such as "Linux Linux" on line 257, "htbp" in Table 6, "Table table" on line 811, and "Table table" on line 848.

• The presentation of findings/takeaways with a gray background detracts from readability.

**Suitability:**

3

---

### Official Review · Reviewer_NtkD · 2024-05-23

**Rating:** 2
**Confidence:** 4

**Summary:**

The paper describe a lab-based evaluation of video conferencing applications, primarily focusing on Zoom, and a crowdsourcing-based dataset involving paired comparisons of simulated video conferencing clips. The authors obtained metrics on typical bitrate allocation strategies, and observed different priorizations regarding audio, video, and screen sharing content. From the crowdsourcing study, they obtained "QoE" ratings based on the PageRank algorithm, and tested different algorithms to model these scores.

**Strengths:**

The paper addresses an important application that is relevant to many Internet users. It presents directly applicable results such as the strategies used by Zoom; these values (regarding bit rates and adaptation strategies) can help understand how to provision such services. In general the paper is written well (except for some details, see below), and the lab-based approach looks technically sound. I understand this to be quite an effort from an implementation perspective, so I commend the authors on achieving this.

**Limitations:**

My main concern with the paper is that it does not deliver what it seems to promise initially. The paper is a bit imbalanced in the sense that it spends quite a bit of detail on the lab-based study, but leaves out crucial aspects for the crowdsourcing part, and finally adds a QoE evaluation into the last two pages without room for the required details and the required discussion. It talks about three VCAs but only effectively describes one in detail. Hence, my recommendation would be to reconsider the framing of the paper and the goal for submission; perhaps leaving out the QoE modeling altogether could give more room for a better description and analysis of the other two VCA methods and the crowdsourcing results. Right now I am having a hard time even ensuring that the data is plausible, so I recommend resubmitting.

Here are detailed, major comments:

- The paper talks about three VCAs, but only Zoom is evaluated in-depth in the paper (esp. considering Figure 2, which I believe to be the most interesting part), while the other two are only mentioned later on. This is a bit misleading to the reader. It would be better to either evaluate all three VCAs in-depth or to focus on Zoom only and mention the other two where relevant.

- Link and explain SSIM briefly; you mention it without explaining how you calculate it and hwo to interpret it. Normally SSIM values are given in the range [-1, 1], not [-100, 100]. Also, is there a reason that the image-only quality metric SSIM was used when better full-reference video-based metrics like VMAF are available? In particular, I am having troubles interpreting the resulting SSIM values with a resolution as little as 144p. Is that range of 0.70 still considered good? Also, you say it's intriguing that when the framerate nears zero the SSIM value is still high. I would argue that this is not surprising at all, as the SSIM metric is not framerate-dependent. This also renders the SSIM metric useless for the video quality assessment under packet loss.

- I very much like Fig. 2, but it concerns me that the tested ranges do not extend to an unlimited bandwidth on the x-axis. This would help to understand better where the threshold is, and in fact we cannot rule out that even above 1 Mbit/s there is a bandwidth-dependent adaptation going on.

- Regarding the method used for comparison: Ref 23 is simply "Standardization Sector and OF Itu. 2013. ITU-T." – what is this reference specifically? Please fix the citation. If you mean DSCS, please cite the most recent version if ITU-T Rec. P.910, not the 2013 version. But bear in mind that DSCS does not let users explcitily compare the Quality of Experience (as stated in your manuscript), but the *impairment* of the second stimulus. This is a crucial difference! Normally, when asking for QoE ratings, an ACR-type scale is used. It depends a bit on what you asked the users, so please clarify this aspect. In fact we do not get a lot of information on the actual test procedure in the paper.

- You do not mention any kind of data cleansing for the user ratings, minimum hardware/software conditions, trick questions, or other ways to ensure the quality of crowdsourced data. This is a crucial step in any crowdsourcing study, and it is important to mention it in the manuscript, since a significant portion of user ratings may be unreliable when conducted in such a fashion. Furthermore, you do not detail how you actually rendered/delivered the media content to the users. This is important for the reproducibility of your study and to ensure no further biases were introduced.

- The choice of the PageRank algorithm in this paper is ... interesting. I have not previously seen it used for QoE prediction. The description of the algorithm is only linked from the original paper, but it would be better to actually show the calculation in your manuscript. More importantly, one would usually use the Bradley Terry model to reconstruct single-valued scores for a set of paired comparisons. Please explain why you chose PageRank over this method. Also explain the absolute values of "QoE" that you obtained, as the range is not clear from the text, and even differs among the scenarios (e.g. in Figure 4), since it is . This would mean the QoE values are not directly comparable between scenarios, and hardly usable.

- Section 5.2 and 6 are quite short and lack context. It almost seems like they were added to the paper on short notice without proper review. For example, you say "we set 𝑡𝑜𝑙 = 10𝑒 − 6, 𝑟𝑎𝑛𝑑𝑜𝑚_𝑠𝑡𝑎𝑡𝑒 = 0, and 𝑠𝑜𝑙𝑣𝑒𝑟 ="𝑛𝑒𝑤𝑡𝑜𝑛 − 𝑐h𝑜𝑙𝑒𝑠𝑘𝑦";" — where did you set that? There is no mention of a particular software or algorithm used. Similarly, the parameters for the other methods may not be specified in sufficient detail to allow implementing the method. Then, in 6.1., you talk about a "SequenceMatcher" without this appearing anywhere else in the paper. In Figure 4 you show a single blue line "Our Model" but it is not clear which of the four algorithms was used here.

- The introductory parts claim that the VCA applications have room for improvement in terms of QoE, and that the results may help in finding more optimal allocation strategies for shared Internet access scenarios, but from the actual results/findings I do not see such a clear conclusion. The measured KPIs are left to be interpreted by the reader, as there is no consensus on how to provide a joint QoE score for interactive VCA sessions. This is exacerbated by the fact that for video, only SSIM was evaluated, which leaves out the very important factor of motion. I would not make such claims without a more rigorous evaluation of the QoE. Whether the crowdsourced study is sufficient for this is debatable, as we lack details on its conduction. Also, in the paper, we do not see any further discussion of the "joint allocation" aspect.

- The paper is missing a discussion section that would tie the results together and provide a more in-depth analysis of the findings, also highlighting possible limitations and drawbacks. This would be crucial for the reader to understand the implications of the results and in which context they can be used.

- You talk about supplementary material but I have seen none. This is a crucial part of the paper, as it would allow the reader to reproduce your results. Please provide this material.

Some further minor comments:

- Spell out and define the terms when first used: QoE, IRB
- "innovative frameworks" - clarify what is innovative about them
- capizatlization: "packet Loss"
- What is a "measurement-driven functional model"?
- It would be good to go into more details on the "QR code recognition". I have an understanding of how this works from other papers/methods, but it is not sufficiently explained here.
- Typesetting: spacing between "audio >screen >video"
- Table 2, 3, 4 and 5 overflow the column width. This could be fixed by using a smaller font size or transposing the table.
- You claim "Users often seek a balance between clarity and fluidity, preferring not to experience significant lags for the sake of sharpness." This is a very broad statement and should be backed up by references.
- Spelling: "Here’s" and typo: "namely𝑢", "use ratings", "Obtain All user ratings"
- Link the "scenarios" you list in Section 5.2 back to Section 3.2 so that the reader can easily understand what you are talking about.
- "Table table 9,"

**Suitability:**

3

---

### Official Review · Reviewer_Ye81 · 2024-05-24

**Rating:** 6
**Confidence:** 4

**Summary:**

This paper provides a detailed study of bandwidth allocation across different media sources such as audio, video, and screen sharing in video conferencing, and their impact on QoE. For this, three different video conferencing applications, namely, Zoom, Webex, and Google Meet are considered. A well-designed subjective study followed by a well-defined model development process has been done in this study.

**Strengths:**

Strengths
1) A systematic approach has been followed to tackle the defined problem.
2) A relatively large-scale user study with 800 participants has been carried out.
3) The dataset will be made public.
4) Using the page-rank approach to develop the QoE model is very interesting.
5) The scenarios are well-defined.
6) The case study on Zoom is very detailed and presented well and provides interesting insights

**Limitations:**

Weaknesses
1) The subject distribution is skewed in terms of the countries they come from with 80% of them being either from the USA or India. This may have an impact on the overall generalizability of the results and the QoE model.
2) Was any metric like POLQA or PESQ used to quantify audio quality like SSIM being used for video quality quantification?

Minor editorial changes
1) Spell out IRB once in the beginning
2) Line 257: "Linux" is used twice
3) Line 604: "2 FPS" --> "3 FPS"
4) Line 587, column 2: "use" --> "user"
5) Line 848: "Table" is used twice

**Suitability:**

3

---

### Official Review · Reviewer_7U4A · 2024-05-25

**Rating:** 5
**Confidence:** 3

**Summary:**

This paper presents an in-depth analysis of bandwidth allocation strategies among Zoom, WebEx and Google Meet, with specific focus on QoE. To this end, a dataset on 800 participants is collected through Mechanical Turk. In addition, a QoE model is proposed and evaluated.

**Strengths:**

- The paper is well-written, logically structured and easy to follow. The contributions are very valuable, in scope and timely.
- Authors are proposing a very relevant and original view on the given topic, and the proposed model and results offer interesting insights.
- It is appreciated that the authors will make their dataset publicly available upon acceptance.

**Limitations:**

- The related work on QoE Modelling is very short. It would be good to add one or two sentences for each mentioned study to briefly mention the findings/results
- In Table 2, the authors are using SSIM as a quality metric. However, no motivation is given on the choice of this particular metric. In my experience, correlation of SSIM to subjective metrics is limited in general and highly dependent on the particular use case. Especially w.r.t. to QoE, VMAF or VQM might be better fits.
- In addition, authors claim in Table 2 that SSIM only shows "a little dip". However, the compete range of subjective perceptual quality typically maps to a very narrow portion of the SSIM scale (typically around 80-90), such that a difference of 5 in SSIM can actually be considered rather big.
- Although I understand and respect the choice to adhere to ITU Recommendations, I am wondering whether in this particular case a Single Stimulus could have been argued. It would have reduced the need for "the accumulated score" as presented by the authors as well as the required PageRank algorithm (as a straightforward MOS would have been possible). Moreover, as the choice was made to offer the two sequences of the Double Stimulus sequentially to the users rather than concurrently, memory bias (even on this short timescale) may bias ratings. It might be good for the authors to briefly discuss this, or to mention it as part of the limitations.
- Was there a Hidden-Reference included?
- While the authors are putting a lot of focus on QoE, i.e. the subjective degree or annoyance, there is also the aspect of ability, i.e. the extent to which users are still able to conduct video conferencing (communicate, give presentations etc.) under limited QoE. For the latter, boundaries can typically be stretched much further. Can the authors give their take on this?
- Section 6: How were the parameters of the presented modelling approaches determined? In case they were tuned during cross-validation, an additional test set should be split of for unbiased evaluation rather than reporting CV results.
- Are the CV folds determined at user or video level? In case of the former, doesn't it include bias if the same video, graded by different users, appears in both training and test set? How would the model perform on a previously unseen sequence?
- Why were there no correlation values such as PLCC or SROCC included in the evaluation? It would provide additional insights on the linearity and monotonicity of the generated predictions.

MINOR COMMENTS:
- In Section 2, it would be good to mention the authors of each referred work, e.g. "Chang et al. [5] highlight comparative results" instead of "[5] highlights comparative results"
- Line 257: "Linux Linux TC" --> "Linux TC"
- The gray boxes with findings and takeaways are rather uncommon, and take up quite some valuable space with limited added value given their redundancy. Authors might consider removing them to create additional space for addressing other issues that require additional explanation.
- Line 643: "namelyu_i,i+1" --> "namely u_i,i+1" (add space)

**Suitability:**

3

---

### Meta-Review · Area_Chair_wfZS · 2024-06-30

**Recommendation:** Accept (Oral)
**Confidence:** 4

**Metareview:**

This paper presents a lab-based evaluation of video conferencing applications with focus on QoE parameters. The authors obtained metrics on typical bitrate allocation strategies, and observed different priorizations regarding audio, video, and screen sharing content. From the crowdsourcing study, they obtained quality ratings based on the PageRank algorithm, and tested different algorithms to model these scores. The paper is well written and highly relevant to the conference. All the reviewers agree on the potential, interest and quality of the approach. The authors provided a very thorough review that led to less positive reviewers to update their assessment towards accept. I agree that the paper should be accepted as oral presentation, given that the authors take care of the risen points by the reviewers.